# Colonoscopy Colorectal Cancer Screening Programme in Southern Iraq: Challenges, Knowledge Gaps and Future Potential

**DOI:** 10.3390/jpm13020173

**Published:** 2023-01-19

**Authors:** Laith Alrubaiy, Ali Al-Rubaye, Wisam Alrudainy, Mazen H. Al-Hawaz, Raja A. Mahmoud, Brian P. Saunders

**Affiliations:** 1Gastroenterology Department, Swansea University Medical School, Swansea SA8 2PP, UK; 2Medical Research Unit, Basra Health Directorate, Basra 289, Iraq; 3Iraqi Medical Association, Basra Branch, Basra 289, Iraq; 4Department of Surgery, Al-Zahraa Medical College, Basra 289, Iraq; 5Public Health Department, Al-Zahraa Medical College, Basra 289, Iraq; 6Bowel Cancer Screening for North West London, St Mark’s Hospital, Watford Road, London HA1 3UJ, UK

**Keywords:** endoscopy, colonoscopy, colorectal cancer screening, bowel cancer screening, Middle East, bowel cancer, Iraq

## Abstract

Data on current colorectal cancer screening practices in Iraq are limited. This study aimed to better understand the current colorectal cancer screening practice and perceived barriers. The project also aimed to use UK expertise to introduce Bowel Cancer Screening Programme (BCSP) in Basra, Iraq. The study consisted of two parts: A pre-visit online survey of clinicians to test the project’s feasibility. A public survey was conducted to understand and gauge the general knowledge and perceived barriers to having colorectal cancer screening. The second phase included a short visit to Basra and the delivery of a multidisciplinary meeting for bowel screening colonoscopists. Fifty healthcare providers completed the survey. Basra has no established bowel cancer screening programme, let alone the country. Opportunistic colonoscopy surveillance is done on an ad hoc base. A total of 350 individuals completed the public survey. The survey showed that more than 50% of participants were not familiar with the concept of a BCSP and less than 25% were aware of “red flag” symptoms of bowel cancer. The short visit to Basra included a roundtable discussion and delivered a training workshop for screening colonoscopists using UK training materials in conjunction with the Iraqi Medical Association. Feedback from the course was extremely positive. Several potential barriers were identified to participate in BCSP. The study highlighted potential barriers, including a lack of public awareness and insufficient training resources to be addressed in future screening programmes. The study has identified several potential areas for future collaboration to support the development of a BCSP centre in Basra.

## 1. Introduction

Colorectal cancer (CRC) is a leading cause of cancer-related death in men and women worldwide. Cancer outcomes are unfavourable in developing countries due to limited health resources and political and economic instability [1]. The World Health Organization (WHO), the Iraqi Cancer Board and the Iraqi Ministry of Health regularly publish cancer incidence and mortality rates [2]. The data collected by the Iraqi Cancer Board identified CRC as the seventh most common type of cancer among both male and female Iraqis [2]. The incidence rate increased from 2.75/105 in 2002 to 3.26/105 in 2011. Around half of the cases were diagnosed in the age group of 40–59 years [2,3]. In the UK and Western countries, mortality rates associated with the disease have stabilised or declined. However, the burden of CRC is increasing in most low-income and middle-income countries. This is possibly due to population awareness of, and adherence to, preventive health behaviour, early detection and screening, and referral for treatment [4]. From local experience, patients with CRC in Iraq present late in their disease with multiple metastases when treatment options are limited. Traditions and cultural factors act as barriers to having screening colonoscopy, which most patients perceive as unacceptable [5,6]. Risk factors for CRC can be divided into non-modifiable risk factors (age, individual medical history and family medical history) and modifiable risk factors (such as diet, lack of physical activity, alcohol consumption and smoking).

Many problems have been historically rooted in the Iraqi health system. A previous BSG-funded visit to Iraq identified that there is a lack of national standards and guidelines, a lack of organisation and ineffective resource allocation [7]. Despite the ongoing political division and poor security situation, damage to the health infrastructure and loss of many health professionals who fled the country, there has been significant momentum in Iraq to start cancer screening programmes which started a few years ago. A good example is the breast cancer screening programme.

In this study, we sought to better understand physicians’ current colorectal cancer screening practices in Basra, in southern Iraq. We hypothesised that several barriers prevent widespread colorectal cancer screening and the development of national guidelines. Identification of these barriers is necessary to improve screening and ultimately increase early detection, prevention and prompt treatment of colorectal cancers. This project also aims to support doctors and nurses in setting up and delivering a bowel cancer screening programme in Basra, in southern Iraq, using the UK Bowel Cancer Screening Programme (BCSP) as an example and with the long-term aim of replicating this centre across the country. In collaboration with the international section of the BSG [8], the project would provide ongoing mentorship to set up a BCSP centre in Basra. The project would also offer public workshops and seminars to increase awareness of the disease risk factors, red flag symptoms and the importance of early detection and screening for CRC.

### Aims and Objectives

The project aimed to better understand current colorectal cancer screening practice and barriers to adoption in Basra. This project would help to promote the development of a local bowel cancer screening centre in Basra capable of delivering and leading a bowel cancer screening programme supported by local healthcare professionals

## 2. Methods

### 2.1. Pre-Visit Activities: (Online Surveys)

#### Assessing the Current State of Bowel Cancer Screening in Basra

Part of the project proposal was to pilot a pre-visit online survey to test the project’s feasibility and provide a guide to the range, extent and level of organisation of endoscopy services in Basra. A Survey Monkey tool was used to develop and devise the questionnaire. Gastroenterologists, general physicians, GI surgeons and nurses involved in managing patients with CRC in Basra, Iraq, were contacted to inform them of this initiative. They were asked to complete an online survey to ascertain their experience managing CRC patients, their learning needs and the endoscopy service provision in their catchment area. This survey assessed the feasibility of the project, exact training needs and existing infrastructure available to enhance the benefit of the subsequent visit [5,9]. Working with the Iraqi Medical Association (IMA) and the Basra Health Directorate, the survey was sent to Basra’s four main hospitals performing endoscopic procedures and all primary healthcare centres.

### 2.2. Knowledge and Attitude of the Public towards Bowel Cancer Screening

A survey included questions related to CRC symptoms, awareness of screening tools, barriers to screening and understanding of different risk factors [5]. A public survey was conducted to understand and gauge the general knowledge of bowel cancer, barriers to having a colonoscopy, and common myths and concepts of bowel cancer. We used a paper-based questionnaire to collect anonymised data on demographic variables, including sex, age, highest education level obtained, marital status, employment, family history of CRC or having a friend with CRC. We also gathered information to assess the attitudes and behaviours of participants and evaluated the knowledge and intent to undergo CRC screening and perceived barriers to CRC screening.

We obtained the project’s required ethical and governance approval from the Medical Research Unit in Basra. The project was approved as a clinical service evaluation/improvement study. No identifiable patient information was shared with a third party. The survey was disseminated through various primary health care centres to capture a representative sample from different areas of Basra. During the survey, it was ensured that participants independently filled out the questionnaire with the option to withdraw from the survey if they so wished. Patients’ care was not affected in any way.

### 2.3. Development of Strategic Aims for the Visit

The pre-visit surveys provided a platform to plan a short-term visit to Basra to carry out the project. However, due to the COVID-19 pandemic and travel restrictions, not all of the UK panel could travel to Iraq. Instead, they provided guidance and oversaw the project. Online materials and resources from the Bowel Cancer Screening unit at St Mark’s Hospital were used in the workshop.

### 2.4. 5-Day Visit to Basra 20–24/9/2021

Formal engagements were arranged during the early part of the visit so that the project team could better understand the training and healthcare systems operating in Basra. This allowed the team to understand the context and pressures of gastroenterology and endoscopy practice in day-to-day healthcare in Basra to appreciate resource constraints and differences in training pathways and standards compared to the operating systems in the UK.

### 2.5. Delivery of a One-Day Multidisciplinary Workshop on the Barriers and Facilitator of the Bowel Cancer Screening Programme in Basra

The one-day conference provided a multidisciplinary open discussion on the feasibility and the need to set up a bowel cancer screening programme in Basra. Attendees included surgeons, gastroenterologists, oncologists, biochemists, general practitioners and other healthcare providers. The discussions covered the themes raised in the pre-visit survey.

Recorded lectures and talks from the Bowel cancer screening programme teams from St Mark’s Hospital were used. UK-JAG colonoscopy standards and BCSP were used as the standards of care. The session included case studies and explored the future role of nurse endoscopists/specialists in delivering BCSP in Basra. However, the lack of appropriate funding and training was identified as a major barrier in the current situation.

The workshop raised public awareness through televised interviews with the team on local and national media channels. Leaflets on bowel cancer screening programs and demonstrations of how to use faecal occult blood testing (FOB) were discussed and approved for dissemination.

### 2.6. Post-Visit Activities

Since the visit, we have disseminated information leaflets and organised public awareness sessions to encourage wider engagement with BCSP. We continued to collect feedback. The team has been invited to present at the joint Iraqi–UK medical association meeting in 2022 to share the project and recommendations. The team is also working with local healthcare professionals to develop guidelines and patient information leaflets.

## 3. Results

### 3.1. The Current State of Bowel Cancer Screening in Basra

Fifty healthcare providers completed the survey. Basra has no established bowel cancer screening programme. Stool testing for occult blood based on the guaiac stool test is available, but FIT stool testing has only recently been introduced. Opportunistic colonoscopy surveillance is carried out on an ad hoc basis. Four major hospitals in Basra provide diagnostic upper and lower GI endoscopy. There were no agreed performance standards on a local or national level. Hospitals had good access to modern endoscopic equipment, accessories and diathermy. Around 30% of respondents reported using a structured referral form for endoscopy. Written information leaflets about procedures are limited. However, most endoscopists (95%) said they are taking consent, but the practice of informed consent falls short of UK standards. Centres with many trainees tend to have better experience and awareness of bowel cancer screening. Attendees suggested using the age of 40 years to start the BCSP as they reported that cancer was seen at a younger age than in the West. With the Iraqi government’s initiative to offer health insurance, individuals will be able to have stool FOB tests and investigations in the private sector giving better access to investigations and shorter waiting times. CT virtual colonoscopy is not a wellestablished modality to screen for bowel cancer; radiologists do not perform this test routinely.

### 3.2. Knowledge and Attitude of the Public towards Bowel Cancer Screening

In all, 350 individuals completed the survey (Table 1). There were participants from different suburbs of Basra. Most of the respondents were males and the mean age of the participants was 40 years. Most of the respondents had completed secondary or university degrees. More than 60% of participants were employed. Most participants did not report a family history of bowel cancer. Although the majority (70%) knew that colon cancer could be fatal, half of the individuals were not aware of colon cancer screening tests. General knowledge about colorectal cancer was low; less than half of the respondents answered the CRC knowledge questions correctly (Table 2).

The survey showed that more than 50% of participants did not know about BCSP and less than 25% were aware of bowel cancer’s red flag symptoms. Individuals reported that their main source of information was family and friends (15%) and information leaflets given in primary care centres (30%). Only 10% reported that they would take a screening test for CRC if their doctors advised doing so (Table 3).

We distributed information leaflets and posters in pharmacies and primary health care centres to distribute to the public. We offered free stool FIT kits for individuals over the age of 50 years. The local GI hospital offered a free colonoscopy test if the FIT was positive. However, no patients were willing to take part in BCSP. However, most patients were grateful for the information leaflets, especially knowing about the red flag symptoms to report to their doctor. Several potential barriers were identified for participation in BCSP. The answers for not taking part in bowel cancer screening:Financial cost—even if colonoscopy is free, you still need to take the day off and pay for travel;Work commitments;Travel/distance;Embarrassment;Faith—cannot change fate;Other priorities in life are caring for family;Even if I have cancer, I do not want to know;Colonoscopy is painful.

## 4. Discussion

The project showed numerous challenges in setting up a bowel cancer screening programme in Iraq. Although physicians and healthcare providers perceived the value of bowel cancer screening programmes, a national program has not been undertaken because of Iraq’s unstable political and security situation. There have been several attempts to set up a national programme, but the regional challenges limit the setting up and rolling out of such a programme [7,9].

Public knowledge and engagement are pivotal in sustainability and success for such a programme to be effective [5,10]. Our project has shown that the general understanding and awareness of bowel cancer screening are poor. This finding was similarly noted in nearby countries [6,11,12,13,14]. The main purpose of this research is to enhance the participants’ knowledge regarding bowel cancer. If people are not given proper education, they remain unaware of the importance and availability of bowel cancer screening tests.

Basra is an oil-rich city in southern Iraq, yet current political and economic instability has had a major impact on the development of health services [9]. There is, however, an opportunity to set up a successful screening programme [15,16]. Knowledge and motivation are important for a successful programme with good public uptake [17,18,19]. We have shown that working collaboratively with local teams can help to bring positive outcomes. Working in collaboration can be carried out by highlighting the project’s main goals among the participants.

It is very important to highlight the limitations of the research and analyse how its boundaries can be overcome. The study sample size is small, and this research is only based on the target population who had attended the centres compared to the local visits. An appropriate sample of participants were invited to take part of the survey. The majority of the survey was carried out in Basra’s city center due to the availability of research professionals. When attempting to generalize the results, this may explain the high proportion of participants (113/350) having a university degree. This may have led to selection bias. However, the data provide good insight into the current situation and potential barriers to setting up BCSP in the region.

### 4.1. We Proposed the Following Future Plans

#### 4.1.1. Adoption of a Regional Approach to Set Up BCSP in Basra

It may be practical to work with the GI teams in Basra to agree on using an agreed format and set of documentation relating to BCSP. Meetings with the local directors and officials agreed on the potential benefits. Further work would be needed to initially assist in producing a paper-based documentation format based on the BCSP standards in the UK. Additional contact and support for the units taking this forward would be needed, and a future visit remains possible. BCSP will be adopted more widely across Iraq if we show evidence from the Basra region that it is possible to promote improvement in cancer detection early.

#### 4.1.2. Adopting Online Training Resources

Access to online materials is not a problem in Iraq. This provides scope for a collaborative training project to improve the skills of trainee endoscopists. Colleagues from Iraq were invited to the Frontiers meeting of St Mark’s Hospital and other online resources to support training [20].

#### 4.1.3. Simulation Training

Although simulation training has shown improvement in endoscopy training, endoscopy training in Iraq is still based on a hands-on basis. There is a desire for an endoscopy skills laboratory to assist movement, but this would require a significant government investment of time and money to make this sustainable [21].

#### 4.1.4. Research Collaboration

The visit did provide some opportunities to collaborate in research, particularly in the epidemiology of cancer, risk factors, patient behaviours and attitudes, endoscopy training and documentation of endoscopy training and the possible use of an electronic reporting system.

#### 4.1.5. Arranging Training Opportunities for Individuals

Interest was expressed in setting up specific training opportunities for individual endoscopy trainees to access UK-style training courses as part of scholarships awarded by the Iraqi government. We will put this proposal forward to the BSG and St. Mark’s Academic Institute to explore the logistics and feasibility of this.

## 5. Conclusions

The main purpose of this research includes providing information and workshops to help and develop a bowel cancer screening programme in Basra with a vision to establish a “Bowel Cancer Screening Centre” in Basra. The screening procedure is also related to motivating and supporting candidates to develop local/national bowel cancer screening standards and referral protocols based on the BSG guidelines. This project has raised public awareness and understanding of the importance of screening for bowel cancer and reporting red-flag symptoms.

## Figures and Tables

**Table 1 jpm-13-00173-t001:** Demographics of individuals completing the public knowledge questionnaire on CRC.

Variables	Number (Total Number of Individuals Screened Are 350)
<50 years	210
>50 years	140
Males	250
Females	100
Education level:	
1. No school education	50
2. Primary school	87
3. Secondary school	100
4. University or above	113
Not married	98
Married	252
Not working	110
Working	240
Family history of colon cancer	
Yes	56
No	294

**Table 2 jpm-13-00173-t002:** Percentages of correct answers for public knowledge of colorectal cancer.

Questions	Percentages of Yes/Correct Answers
General knowledge questions	
Have you ever heard of early colon cancer screening tests?	50%
Colon cancer is potentially a preventable condition	10%
I think colon cancer can be fatal	70%
Colon cancer recovery rates increase when detected in the early stages	10%
Men and women are prone to colon cancer	50%
Symptoms of colon cancer	
The presence of blood in the stool	25%
Recent unintentional weight loss	10%
New change in the number of bowel motions or habit	23%
I will undergo early screening for colon cancer if my doctor advises me to do so.	10%

**Table 3 jpm-13-00173-t003:** Sources of information about bowel cancer screening programme (if you have heard of it) N = 175 (50% of total number).

Sources of Information	Percentages of Patients Using the Information Sources
Family and friends	25%
Information booklets	30%
Radio/TVC	5%
Newspapers	10%
Doctors or healthcare providers	20%
Twitter/Facebook/WhatsApp	10%

## Data Availability

The data generated in the present study may be requested from the corresponding author.

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
