# Peer review of "Colonoscopy Colorectal Cancer Screening Programme in Southern Iraq: Challenges, Knowledge Gaps and Future Potential"

_jpm, 2023, doi:10.3390/jpm13020173_

Round 1

Reviewer 1 Report

In this manuscript, the authors shared the results of their pre-visit surveys as well as the on-site activities to promote BCSP in Basra, Iraq. As pointed out by the authors, the current study sample size is relatively small. However, their results demonstrated the current situation of the colorectal cancer screening practice and the knowledge of the public in Basra, which is of great importance for the further implementation of the project. I think this paper will be of interest to many clinicians and researchers as well as the general public, and look forward to seeing the updated research results regarding this project in the future.

Author Response

Dear respected reviewer,

On behalf of my co-authors, I would like to thank you for reviewing our manuscript. We also greatly appreciate your complimentary comments and thoughtful suggestions that add substantial depth and impact to our findings.

Best regards

Dr Laith Alrubaiy

Reviewer 2 Report

Thank you for the opportunity to review this manuscript. It provides an interesting insight into the challenges for cancer screening in a unique part of the world. I congratulate the authors for their efforts and wish them future success. As a brief report is is well written and informative. My only comment is that the authors could elaborate more about how the public survey was disseminated - were the respondents generally from a more affluent area since many had university education? What was the uptake rate of the survey (350 respondents out of how many survey forms distributed). Was there a potential the result was not reflective of the iraqi population as a whole?

Author Response

Dear respected reviewer,

On behalf of my co-authors, I would like to thank you for reviewing our manuscript and for your thoughtful suggestions that add substantial depth and impact to our findings.

Please find below point-by-point responses to the reviewer’s concerns.  We have made changes ( marked in red colour) in the manuscript to incorporate the reviewers’ comments.  We hope that you find our responses satisfactory and that the manuscript is now acceptable for publication.

Best regards

Dr Laith Alrubaiy

My only comment is that the authors could elaborate more about how the public survey was disseminated - were the respondents generally from a more affluent area since many had university education? What was the uptake rate of the survey (350 respondents out of how many survey forms were distributed). Was there a potential the result was not reflective of the Iraqi population as a whole?

Response: The point raised by the respected reviewer is very valid. A convenience sample of participants was invited to take part in the survey. Most of the survey was carried out in Basra's city centre due to the availability of research professionals. When attempting to generalise the results, this may explain the high proportion of participants (113/350) having a university degree and any potential selection bias. This potential drawback was acknowledged in the manuscript's discussion section.